# Intestinal lysozyme1 deficiency alters microbiota composition and impacts host metabolism through the emergence of NAD+-secreting *ASTB Qing110* bacteria

Chengye Zhang,[1] Chen Xiang,[1] Kaichen Zhou,[2,3] Xingchen Liu,[1,4] Guofeng Qiao,[1] Yabo Zhao,[1] Kemeng Dong,[1,4] Ke Sun,[1] Zhihua Liu[1,4]

**ABSTRACT**   The intestine plays a pivotal role in nutrient absorption and host defense against pathogens, orchestrated in part by antimicrobial peptides secreted by Paneth cells. Among these peptides, lysozyme has multifaceted functions beyond its bactericidal activity. Here, we uncover the intricate relationship between intestinal lysozyme, the gut microbiota, and host metabolism. Lysozyme deficiency in mice led to altered body weight, energy expenditure, and substrate utilization, particularly on a high-fat diet. Interestingly, these metabolic benefits were linked to changes in the gut microbiota composition. Cohousing experiments revealed that the metabolic effects of lysozyme deficiency were microbiota-dependent. 16S rDNA sequencing highlighted differences in microbial communities, with *ASTB_g* (OTU60) highly enriched in lysozyme knockout mice. Subsequently, a novel bacterium, *ASTB Qing110*, corresponding to *ASTB_g* (OTU60), was isolated. Metabolomic analysis revealed that *ASTB Qing110* secreted high levels of NAD+, potentially influencing host metabolism. This study sheds light on the complex interplay between intestinal lysozyme, the gut microbiota, and host metabolism, uncovering the potential role of *ASTB Qing110* as a key player in modulating metabolic outcomes.

**IMPORTANCE**   The impact of intestinal lumen lysozyme on intestinal health is complex, arising from its multifaceted interactions with the gut microbiota. Lysozyme can both mitigate and worsen certain health conditions, varying with different scenarios. This underscores the necessity of identifying the specific bacterial responses elicited by lysozyme and understanding their molecular foundations. Our research reveals that a deficiency in intestinal lysozyme1 may offer protection against diet-induced obesity by altering bacterial populations. We discovered a strain of bacterium, *ASTB Qing110*, which secretes NAD+ and is predominantly found in lyz1-deficient mice. *Qing110* demonstrates positive effects in both *C. elegans* and mouse models of ataxia telangiectasia. This study sheds light on the intricate role of lysozyme in influencing intestinal health.

**KEYWORDS**   Lyz1, gut microbiota, metabolism, NAD+, aging

The intestine serves a crucial dual function: it absorbs ingested nutrients while also safeguarding the host against invading pathogens. In the large intestine, a robust mucin layer acts as a protective barrier, creating a separation between the densely populated microbial community and the epithelial cells (1). This is in stark contrast to the small intestine, which is coated with a mucosal layer that is both thinner and more permeable. In the small intestine, the primary defense mechanism against pathogens hinges on the secretion of antimicrobial peptides by Paneth cells, a process that is maintained under homeostatic conditions (2).

Address correspondence to Zhihua Liu, zhihualiu@mail.tsinghua.edu.cn.

Chengye Zhang and Chen Xiang contributed equally to this article. Author order was determined by tossing a coin.

The authors declare no conflict of interest.

See the funding table on p. 20.

These antimicrobial peptides encompass Reg3γ, α-defensins, and lysozyme, each playing a unique role in intestinal health. Reg3γ, apart from its bactericidal activity (3–5), also exerts a diverse array of effects on host metabolism, influencing energy balance and the onset of alcohol-induced fatty liver disease (6, 7). α-Defensins contribute to the regulation of intestinal microbial ecology (8), and supplementing human defensin has been shown to ameliorate metabolic dysfunction in a murine model of diet-associated obesity (DIO) (9). In the mouse genome, there are two lysozyme genes: lysozyme M (lysozyme 2) and lysozyme P (lysozyme 1). Lysozyme M is expressed in myeloid cells, while lysozyme1 (Lyz1) is expressed in intestinal Paneth cells (10, 11). Paneth cells, located at the base of intestinal crypts, secrete a significant amount of Lyz1 into the intestinal lumen. Lyz1, being the primary lysozyme in the gut lumen, is also known as intestinal lysozyme. Lysozyme, specifically Lyz1 in mice, influences the intestinal bacterial composition (12). Lyz1 deficiency leads to an increase in muralytic microbes, including *Rumunicoccus gnavus*, which is associated with inflammatory bowel disease. Intriguingly, Lyz1 deficiency has been found to mitigate chemically induced colitis (8). In a seeming paradox, exogenous lysozyme supplementation, whether sourced from hen egg white or the fungus *Acremonium alcalophilum*, has demonstrated a capacity to reduce chemically induced colitis in animal models (13, 14). Moreover, lysozyme from *Acremonium alcalophilum* has been shown to protect mice from microbiota encroachment induced by a high-fat diet and associated fasting hyperinsulinemia (13). These seemingly contradictory effects of lysozyme on intestinal inflammation highlight its complex interactions with microbiota. A comprehensive understanding of its effects necessitates identifying the specific bacteria induced by lysozyme and elucidating the underlying molecular mechanisms.

Nicotinamide adenine dinucleotide (NAD$^+$) plays a pivotal role in numerous biological processes across different organisms. As a co-enzyme in redox reactions, NAD$^+$ accepts protons during glycolysis, the tricarboxylic acid cycle, and oxidative phosphorylation in the mitochondria (15). NAD$^+$ is indispensable as a co-factor or substrate for NAD$^+$-dependent enzymes, such as sirtuins, poly (ADP-ribose) polymerases (PARPs), and NAD$^+$ glycohydrolases (15). Through these functions, NAD$^+$ modulates critical cellular activities, including metabolic pathways, redox homeostasis, DNA repair, epigenetic modifications, chromatin remodeling, immune activation, and axonal degeneration (15). These processes are vital for maintaining systemic homeostasis and overall health.

A decline in NAD$^+$ levels is associated with aging and various diseases. Research has been shown that restoring NAD$^+$ levels can potentially slow down, or even reverse, many age-related diseases (16, 17). As a result, targeting NAD$^+$ metabolism has emerged as a promising therapeutic strategy to enhance health in later life, potentially extending both health span and lifespan.

Research into NAD$^+$ precursors, such as NMN or NR, has garnered attention for their capability to boost NAD$^+$ levels and deliver health benefits (18–21). An emerging and fascinating area of research is the potential role of microbes in modulating host metabolism through their involvement in NAD$^+$ metabolism. It is important to note that NAD$^+$ is inherently unstable and cannot be directly absorbed by the intestine (22). In our investigation of metabolic improvements associated with lysozyme deficiency, we have discovered a novel species of bacteria capable of secreting NAD$^+$. This bacterium has shown beneficial effects in both *C. elegans* and a mouse model of ataxia telangiectasia.

## RESULTS

### *Lyz1*$^{-/-}$ mice displayed an enhanced metabolic profile compared to WT controls

To investigate the impact of Lyz1 on host metabolism, we monitored the body weight of WT and *Lyz1*$^{-/-}$ mice post-weaning (Fig. 1a). Initial weights were comparable between both groups until week 7, when *Lyz1*$^{-/-}$ mice began to exhibit less weight gain than their WT counterparts (Fig. 1b), suggesting a potential metabolic alteration.

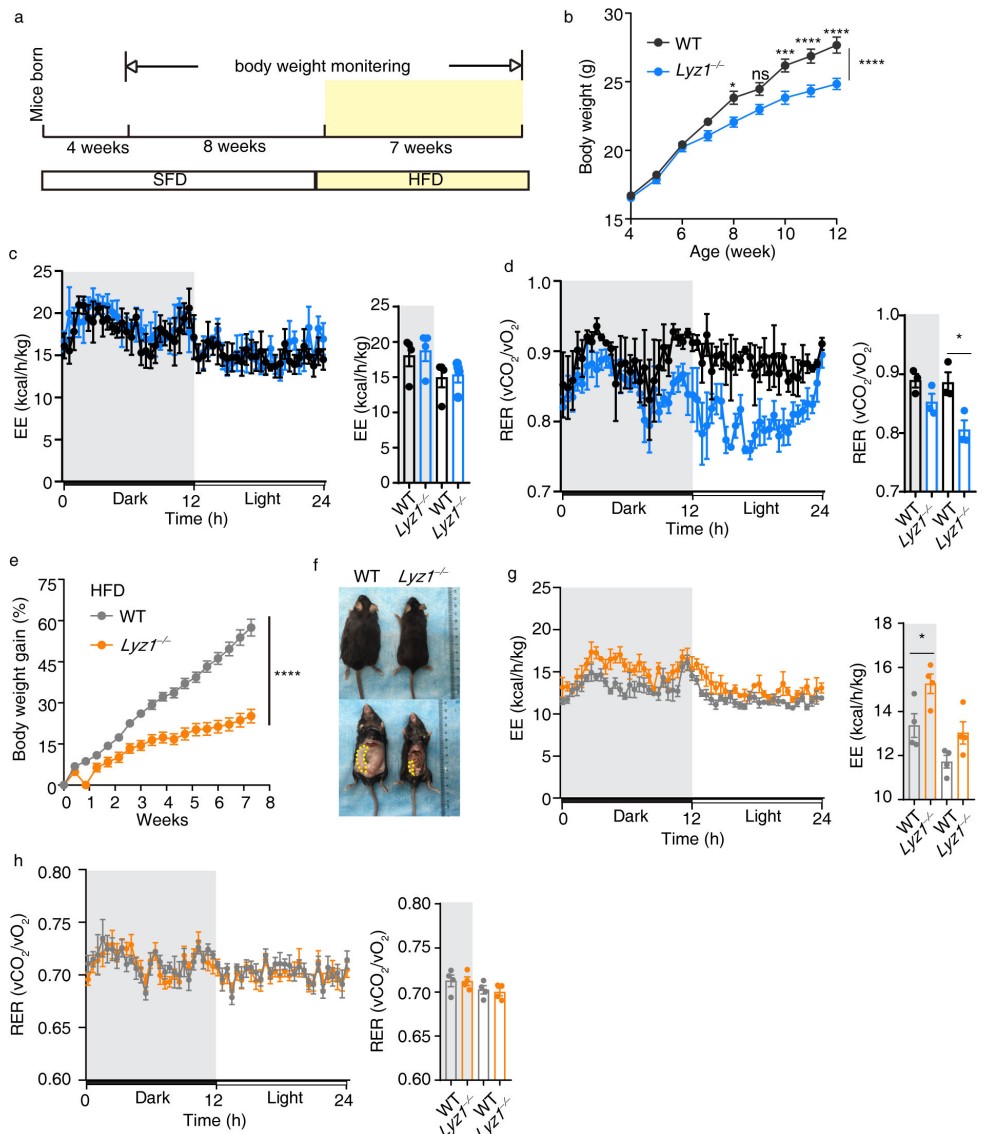

**FIG 1** Improved metabolic phenotype of *Lyz1⁻/⁻* mice on different diets compared to WT controls. (a) Schematic representation of the feeding scheme. (b) Weekly monitoring of body weight from the 4th week after birth for *Lyz1⁻/⁻* and WT control mice (*n* = 9 ~ 10 mice per group). (c) 10-week-old mice's energy expenditure plotted over a 24-h period. (d) 10-week-old mice's respiratory exchange ratio (RER) plotted over a 24-h period. (e) Body weight measurements were taken every 3 days after transitioning mice to a high-fat diet at 12 weeks of age (*n* = 20 mice per group). (f) Comparison of representative WT (left) and *Lyz1⁻/⁻* (right) mice after 7 weeks of high-fat diet feeding. (g) Energy expenditure of 19-week-old mice on a high-fat diet plotted over a 24-h period. (h) Respiratory exchange ratio of 19-week-old mice on a high-fat diet plotted over a 24-h period. Mean values are presented with error bars indicating the standard error of the mean (SEM) in panels (b, e). In panels (c, d, g, h), individual data points are represented by symbols, with the means (±SEM) displayed. Statistical analysis was conducted using two-way ANOVA with Bonferroni's multiple comparisons test for panels (b, e), and one-way ANOVA with Tukey's multiple comparisons test for panels (c, d, g, h). "NS" indicates no significant difference (*P* > 0.05), while asterisks denote significance levels: **P* < 0.05, ****P* < 0.001, *****P* < 0.0001. The data shown are representative of results obtained from at least three independent experiments.

Further exploration using Oxymax CLAMS (Comprehensive Lab Animal Monitoring System) revealed that while food consumption and activity levels remained consistent across both groups (Fig. S1a through c), energy expenditure profiles on a standard feeding diet (SFD) were similar (Fig. 1c). However, during the light phase, *Lyz1⁻/⁻* mice

showed a lower respiratory exchange ratio (RER), indicating a higher rate of fat oxidation, though this difference was not observed during the dark period (Fig. 1d; Fig. S1d and e).

Upon transitioning to a high-fat diet (HFD) at 12 weeks of age, $Lyz1^{-/-}$ mice demonstrated significantly less weight gain (Fig. 1e) and reduced fat deposition (Fig. 1f), all while maintaining comparable food intake and activity to the WT mice (Fig. S1f through h). On the HFD, $Lyz1^{-/-}$ mice exhibited increased energy expenditure during the dark phase (Fig. 1g). Notably, both WT and $Lyz1^{-/-}$ groups displayed similar RER values around 0.7, indicating that fat was the predominant fuel source when mice were on the HFD (Fig. 1h; Fig. S1i and j).

## Microbiome confers the metabolic benefits observed in $Lyz1^{-/-}$ mice

Intestinal lysozyme has a crucial role in sculpting the microbial landscape, with significant implications for host metabolism (13, 23 ). In addition, it may influence host metabolism by releasing bacterial cell wall components from intestinal microbes, potentially enhancing insulin secretion or sensitivity (24). With these potential mechanisms in mind, we aimed to determine whether the metabolic effects of Lyz1 are dependent on the microbial community composition.

We used the established cohousing method to standardize the microbial communities between WT and $Lyz1^{-/-}$ mice post-weaning (Fig. S2a), transitioning from genotype-based separate housing. In contrast to the separately housed mice, cohoused WT and $Lyz1^{-/-}$ mice displayed no significant weight differences, whether on an SFD (Fig. 2a) or an HFD (Fig. 2d). Food consumption and activity levels remained consistent across both groups (Fig. S2b through d). Furthermore, in the cohoused setting, the previously observed differences in respiratory exchange ratio (RER) or energy expenditure (EE) between WT and $Lyz1^{-/-}$ mice, regardless of diet types, were no longer present (Fig. 2b and c; Fig. 2e and f; Fig. S2e through j).

A thorough metabolic trait comparison between separately housed and cohoused WT and $Lyz1^{-/-}$ mice on the HFD revealed distinct differences. Separately housed $Lyz1^{-/-}$ mice had smaller fat and liver tissues compared to WT mice on the HFD (Fig. 2g), a disparity eliminated by cohousing (Fig. 2g). EchoMRI body composition analysis further confirmed that separately housed $Lyz1^{-/-}$ mice had reduced fat content, a difference abolished by cohousing (Fig. 2h).

Examining various adipose tissues, perigonadal visceral adipose tissues (pgVAT), inguinal subcutaneous adipose tissues (ingSAT), brown adipose tissues (BAT), and liver tissues in both housing conditions revealed that adipocytes in all adipose tissues from separately housed $Lyz1^{-/-}$ mice were smaller (Fig. 2i through l), with fewer crown-like structures (CLS) compared to WT mice (Fig. 2n) and had reduced triglyceride content in liver tissues (Fig. 2m). These metabolic advantages were nullified by cohousing, as indicated by the size of adipocytes, CLS number, and liver triglyceride content in WT and $Lyz1^{-/-}$ mice (Fig. 2i through n). We also assessed the impact of housing on glucose tolerance and insulin sensitivity. On the SFD, both separately housed and cohoused $Lyz1^{-/-}$ mice exhibited similar glucose tolerance (Fig. S2k and m) and insulin sensitivity (Fig. S2l and n) as WT mice. However, on the HFD, separately housed $Lyz1^{-/-}$ mice displayed improved glucose tolerance (Fig. 2o) and insulin sensitivity (Fig. 2p), benefits that were eradicated in the cohoused setting (Fig. 2o and p). In conclusion, our data robustly indicate that intestinal lysozyme influences host metabolism in a microbiome-dependent manner.

## *ASTB Qing110* is highly enriched in $Lyz1^{-/-}$ mice

We proceeded to analyze microbial communities in separately housed WT and $Lyz1^{-/-}$ mice using 16S ribosomal RNA gene sequencing. A principal component analysis (PCA) of Bray-Curtis distances revealed notable differences in taxonomic composition between WT and $Lyz1^{-/-}$ mice (analysis of similarities, R = 0.7160, $P$ = 0.0010) (Fig. 3a). Diversity analysis of operational taxonomic units (OTUs) indicated similar community diversity (as measured by Shannon and Simpson indices) but a slightl decrease in community

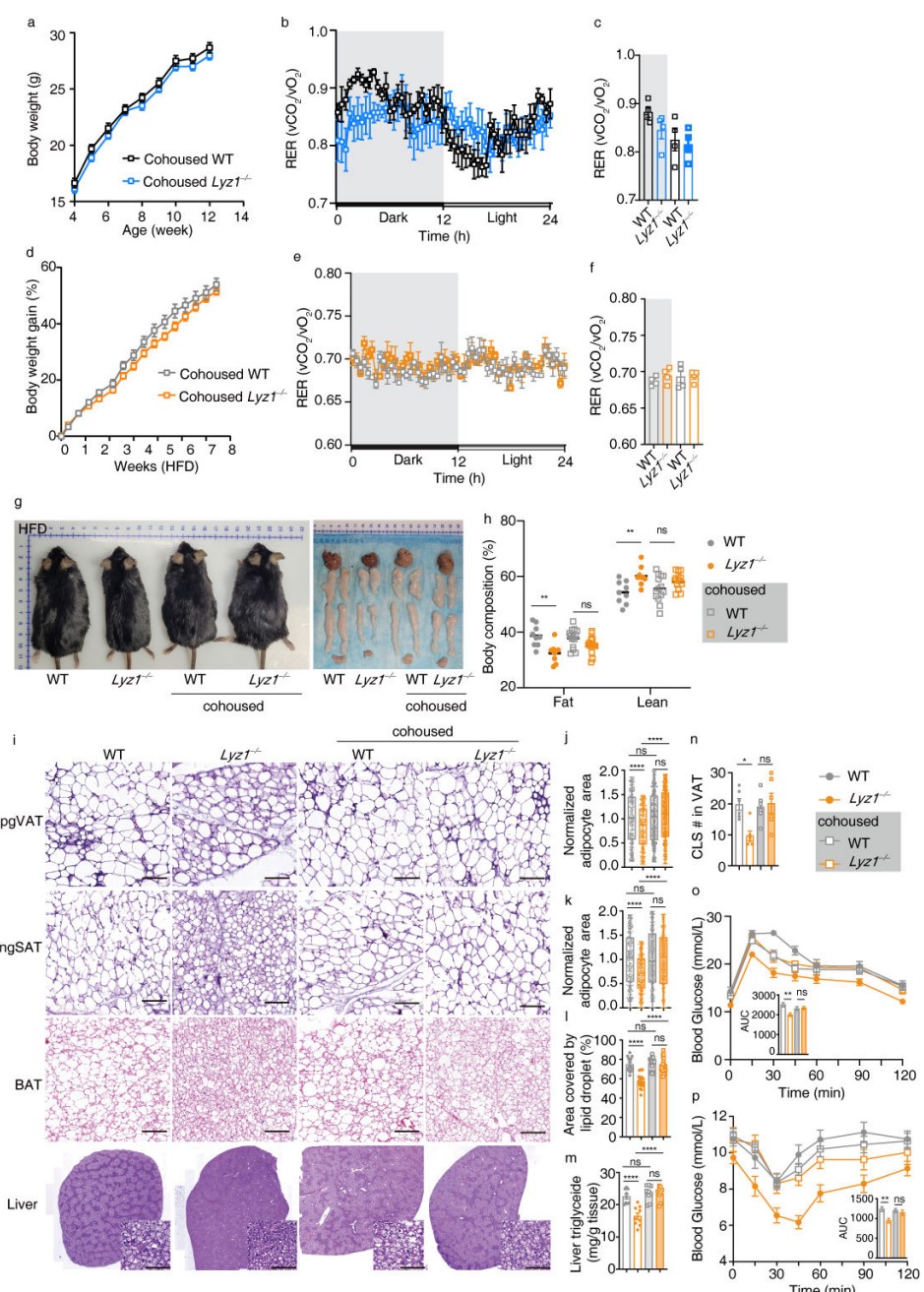

**FIG 2** Impact of cohousing on metabolic benefits of *Lyz1⁻/⁻* mice. (a) Weekly monitoring of body weight from the 4th week after birth for cohoused WT and *Lyz1⁻/⁻* mice (*n* = 14 mice per group). (b) RER of cohoused 10-week-old WT and *Lyz1⁻/⁻* mice plotted over a 24-h period (*n* = 4 mice per group). (c) Average RER of cohoused 10-week-old WT and *Lyz1⁻/⁻* mice during the light and dark periods. (d) Body weight measurements were taken every 3 days after cohoused mice were placed on an HFD at the age of week 12 (*n* = 14 mice per group). (e) RER of cohoused 19-week-old WT and *Lyz1⁻/⁻* mice on HFD plotted over a 24-h period (*n* = 4 mice per group). (f) Average RER of cohoused 19-week-old WT and *Lyz1⁻/⁻* mice on HFD during the light and dark periods. (g) Photographs comparing separately housed and cohoused mice after 7 weeks on HFD. Representative images of the liver, subcutaneous adipose tissue (SAT), visceral adipose tissue (VAT), and brown adipose tissue (BAT) from these groups. (h) Quantitative estimation of fat tissue mass and lean tissue mass evaluated by whole-body EchoMRI analysis of WT and *Lyz1⁻/⁻* mice, separately housed or cohoused, on HFD. (i) Representative H&E-stained images of perigonadal VAT (pgVAT), inguinal SAT (ingSAT), BAT, and liver from WT and *Lyz1⁻/⁻* mice, separately housed or cohoused, on HFD (scale bar = 100 μm). (j) Relative adipocyte area of pgVAT. (k) Relative adipocyte area of ingSAT. (l) Relative lipid droplet covered area in BAT. (m) Levels of triglycerides determined biochemically in the livers of different mice. (n) Number of crown-like structures

**FIG 2 (Continued)**

(CLS) in pgVAT. (o) Plasma glucose concentration and mean area under the curve measured during an intraperitoneal glucose tolerance test (ipGTT) in different groups of mice fed with HFD ($n$ = 7 ~ 9 mice per group). (p) Plasma glucose concentration and mean area under the curve measured during an insulin tolerance test (ITT) in different groups of mice fed with HFD ($n$ = 10 ~ 12 mice per group). Mean values are presented with error bars indicating SEM in panels (a, b, d, e, j, k, o, p). Means (±SEM) are plotted with each symbol representing an individual animal in panels (c, f, h, l-n). Statistical analysis was conducted using one-way ANOVA with Tukey's multiple comparisons test in panels (c, f, j-p) or two-way ANOVA with Tukey's multiple comparisons test in panel (h). "NS" indicates no significant difference ($P$ > 0.05), while asterisks denote significance levels: *$P$ < 0.05, **$P$ < 0.01, ****$P$ < 0.0001. The data presented are representative of results from at least three independent experiments.

richness (as measured by the Ace and Chao indices) in $Lyz1^{-/-}$ mice compared with WT mice (Fig. 3b).

In a detailed comparison of the relative abundance of the top 25 OTUs (Fig. 3c), significant differences were identified for specific OTUs such as $ASTB\_g$ (OTU60), $Allobaculum\_g$ (OTU418), $Clostridium\_g$ (OTU297), $Lachnoclostridium\_g$ (OTU31), and $Muribaculaceae\_f$ (OTU164) between WT and $Lyz1^{-/-}$ mice (Fig. 3d). Notably, $ASTB\_g$ (OTU60) stood out as it was highly abundant in $Lyz1^{-/-}$ mice but virtually absent in WT mice, whereas $Allobaculum\_g$ (OTU418) displayed the opposite pattern (Fig. 3d). Phylogenetically, $ASTB\_g$ (OTU60) and $Allobaculum\_g$ (OTU418) were found to be closely related (Fig. S3a).

Further investigation revealed that the average relative abundance of OTU60 in separately housed $Lyz1^{-/-}$ mice was 15%, which significantly dropped following cohousing with WT mice (Fig. 3e). The abundance of $ASTB\_g$ (OTU60) also negatively correlated with the body weight of the mice (Fig. 3f), aligning with the previously noted observation where cohousing nullified the metabolic benefits observed in $Lyz1^{-/-}$ mice (Fig. 2).

To delve deeper into the bacterium in $ASTB\_g$ (OTU60), bacterium #110 was isolated from $Lyz1^{-/-}$ mice fecal samples. Its 16S rDNA sequence matched 99% with OTU60 ($EF603857\_s$) (Fig. S3b). Phylogenetic analysis placed bacterium #110 close to an $ASTB$ species named $M10-2$, though they shared only 91% sequence identity based on the 16S rDNA sequence, highlighting them as distinct species. Given that $ASTB\_g$ has not been fully characterized, and there has been considerable reorganization within the phylum Firmicutes, we tentatively placed bacterium #110 within the genus of $ASTB$ in the $Erysipelotrichaceae$ family. Bacterium #110 did not correspond to any known bacterial species, sharing just 73% sequence identity with $ASTB$ $M10-2$, as revealed by whole-genome analysis using average nucleotide identity (Fig. S4a). Consequently, bacterium #110 was named $ASTB$ $Qinghuaensis$ $Lab-110$, or $ASTB$ $Qing110$ for short.

We assessed the abundance of $ASTB$ $Qing110$ in $Lyz1^{-/-}$ mice during a high-fat diet regimen. Our findings reveal a significant increase in $Qing110$ levels on day 14, with no notable changes on days 7 and 21. This suggests that $Qing110$'s abundance remains stable throughout high-fat feeding (Fig. 3g). Further exploring the impact of a high-fat diet (HFD) on $Qing110$ colonization, we orally administered $Qing110$ to wild-type (WT) mice on an HFD. The results showed that the average abundance of $Qing110$ was 4.065% (Fig. 3h), indicating that HFD does not impede $Qing110$ colonization.

In investigating the metabolic implications of $ASTB$ $Qing110$, a metabolomic approach was adopted. $Lyz1^{-/-}$ mice had an enrichment of $ASTB$ $Qing110$, whereas WT mice had a higher presence of $Allobaculum$. Including an isolated $Allobaculum$ strain from WT mice in the analysis, notable differences in metabolites between the supernatants of $ASTB$ $Qing110$ and those of the control $Allobaculum$. Remarkably, $ASTB$ $Qing110$ supernatants had elevated levels of NAD$^+$ (Fig. 3i), and its genome analysis unveiled a $de$ $novo$ NAD$^+$ biosynthesis pathway in $ASTB$ $Qing110$ (Fig S3c). HPLC-mass spectrometry analysis confirmed the high levels of NAD$^+$ in the supernatants of $ASTB$ $Qing110$, contrasting the absent NAD$^+$ in the control $Allobaculum$ culture (Fig. 3j). Furthermore, while NAD$^+$ was detected in the cultures of $Longicatena$ $caecimuris$ and $Enterococcus$ $lactis,$ it was not found in others such as $Akkermansia$ $muciniphila$ and $Bacteroides$ $fragilis$ (Fig. 3j).

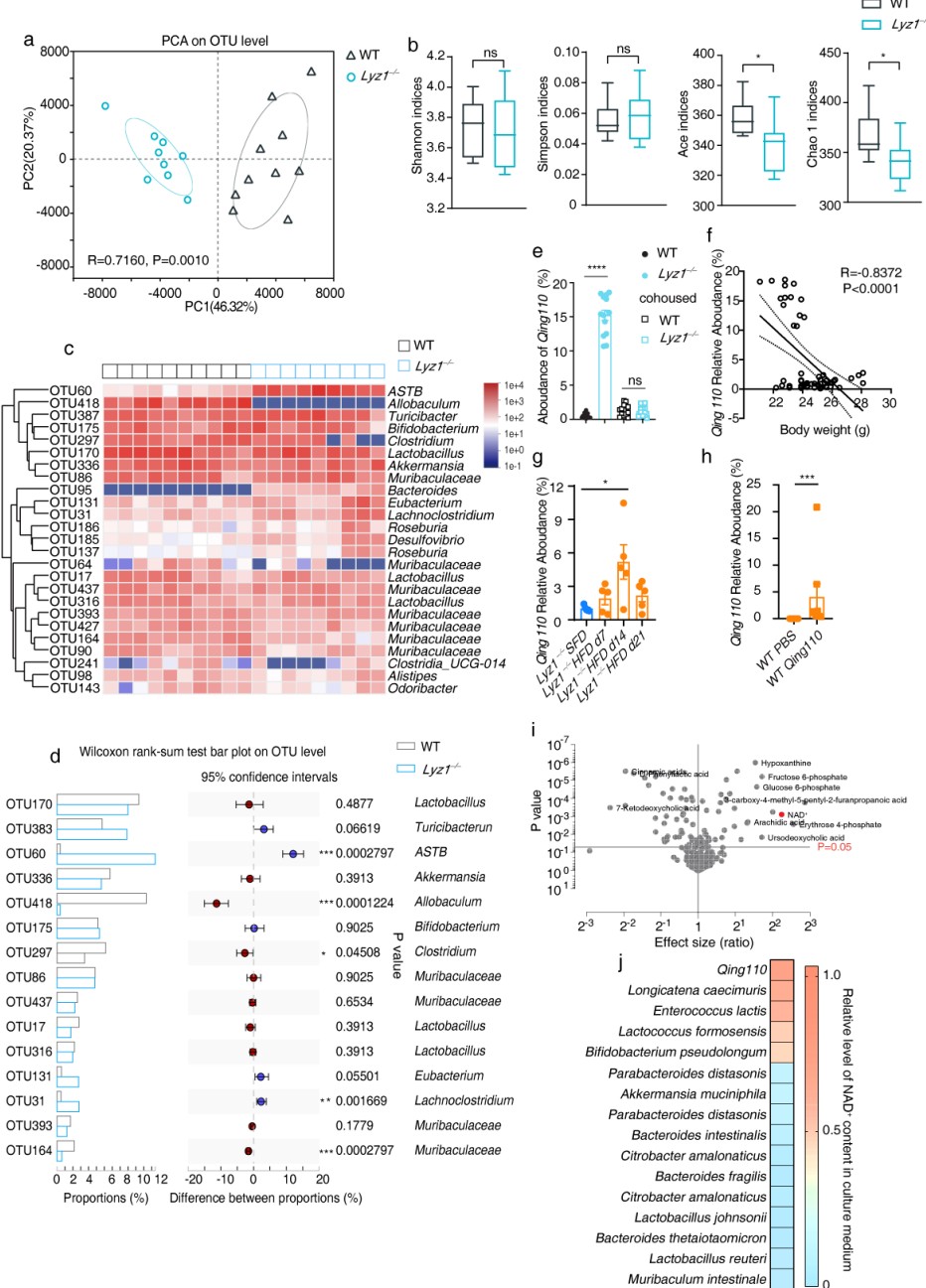

**FIG 3** Differential microbial composition in WT and *Lyz1*⁻/⁻ mice. (a) Weighted UniFrac distance analysis of operational taxonomic units (OTUs) in fecal samples collected from SFD-fed WT and *Lyz1*⁻/⁻ mice, showing separation in gut microbiota composition among the groups based on distance analysis. Each dot represents an individual mouse. (b) Diversity analysis with Shanon index, Simpson index, Ace index, and Chao index of OTUs in fecal samples collected from SFD-fed WT and *Lyz1*⁻/⁻ mice. (c) Heatmap depicting the abundance of the top 25 OTUs in SFD-fed WT and *Lyz1*⁻/⁻ mice. Averages for multiple mice at each group and time point are presented, and the heatmap is based on log10-normalized counts. (d) Relative abundance and statistical description of the top 15 enriched OTUs. (e) Real-time PCR analysis of *Qing110* in fecal samples from WT and *Lyz1*⁻/⁻, separately housed or cohoused. The relative abundance of *Qing110* was quantified by normalizing it to total bacteria. (f) Pearson's correlation analysis between *Qing110* abundance and body weight. (g) Real-time PCR analysis of *Qing110* in fecal samples from *Lyz1*⁻/⁻ mice on SFD and HFD. The relative abundance of *Qing110* was quantified by normalizing it to total bacteria, and the relative abundance of *Qing110* on HFD was normalized to the relative abundance of *Qing110* on SFD. (h) Real-time PCR analysis of *Qing110* in fecal samples from WT mice treated with PBS or *Qing110* for 1 month. The relative abundance of *Qing110* was quantified by normalizing to total bacteria. (i) Analysis of metabolites in supernatants of (Continued on next page)

**FIG 3** (Continued)

*ASTB Qing110* and the control *Allobaculum*. (j) Heatmaps displaying NAD$^+$ levels in culture medium from common intestinal commensal bacteria. The data presented are representative of results from at least three independent experiments. Mean values are presented with error bars indicating SEM in panels (b, d). Means (±SEM) are plotted with each symbol representing an individual animal in panel (e, g, h). Statistical analysis was conducted using two-tailed Student's *t*-tests in panel (b, h), multiple *t*-tests in panel (l), or one-way ANOVA with Tukey's multiple comparisons test in panel (e, g). "NS" indicates no significant difference ($P > 0.05$), while asterisks denote significance levels: *$P < 0.05$, **$P < 0.01$, ***$P < 0.001$, ****$P < 0.0001$.

## *ASTB Qing110* exerts metabolic benefits in the *C. elegans* model

Previous preclinic studies have demonstrated the beneficial effects of NAD$^+$ supplementation against various aging-related diseases in different animal models (25). Given these findings, we aimed to investigate the potential impact of *ASTB Qing110* on the longevity of *C. elegans*, a commonly used model organism in aging research. Initially, worms at the L1 stage were unable to progress to the L4 stage on agar plates containing only *Qing110*. To address this, the worms were cultured on plates with a mixture of *E. coli* OP50 and *Qing110* in a 1:2 ratio, referred to as *Qing110*-containing plates. On these plates, the worms developed normally from L1 to L4. For comparison, NAD$^+$ was supplemented to worms in a separate experiment setup. When worms at the L4 stage were placed on plates with either *Qing110* or NAD$^+$, no change in body length was observed on day 2. However, both *Qing110* and NAD$^+$ led to a significant increase in body length at day 3 (Fig. S5a and b). Neither *Qing110* nor NAD$^+$ supplementation affected the brood size of the worms (Fig S5c and d).

Analysis revealed that worms grown on *Qing110*-containing plates exhibited higher NAD$^+$ abundance levels compared to control worms, with levels comparable to those in worms grown on plates supplemented with 50 µM NAD$^+$ (Fig. 4a). Therefore, worms grown with 50 µM NAD$^+$ were used as the control group in subsequent experiments.

Previous research has established that NAD$^+$ supplementation can extend both the lifespan and health span of worms (18). Consistent with this, worms grown on *Qing110*-containing plates displayed a significant increase in lifespan, comparable to that of worms supplemented with NAD$^+$ (Fig. 4b). Moreover, the *Qing110* treatment enhanced the activity of worms at later life stages (Fig. 4c and d) and conferred heat resistance to a similar extent as NAD$^+$ supplementation (Fig S5e and f). Overall, the data suggest that *Qing110* supplementation has the potential to improve the lifespan and health span of *C. elegans*.

Mitochondrial dysfunction is a well-known characteristic of aging, and increasing NAD$^+$ levels have been shown to enhance both lifespan and health span by boosting mitochondrial activity (18, 26–29). To assess the impact of *Qing110* on mitochondrial function, we monitored mitochondrial content and activity in both young and old worms treated with either *Qing110* or NAD$^+$. Using the zcls14[myo-3::GFP(mit)] strain of *C. elegans*, which expresses a GFP with a mitochondrial targeting sequence under the control of a muscle-specific promoter (30). We observed that *Qing110* and NAD$^+$ supplementation significantly increased mitochondrial content in muscle cells in adult worms at various time points (Fig. 4e through g). In addition, staining with tetramethylrhodamine ethyl ester (TMRE), a dye that accumulates in functional mitochondria, revealed higher levels of mitochondrial activity in worms treated with *Qing110* or NAD$^+$ (Fig. 4h and i), further supporting the positive effects of *Qing110* on mitochondrial function and, by extension, on the health span and lifespan of *C. elegans*.

## *Qing110* supplementation improves metabolic dysfunction and extends lifespan in the ataxia telangiectasia mouse model

Ataxia telangiectasia is a rare inherited autosomal recessive genetic disease characterized by growth retardation, premature aging, sterility, neurodegeneration, insulin resistance, immunodeficiency, and cancer predisposition, particularly lymphomas (31, 32). The loss of the *Atm* gene, which plays a crucial role in DNA damage repair,

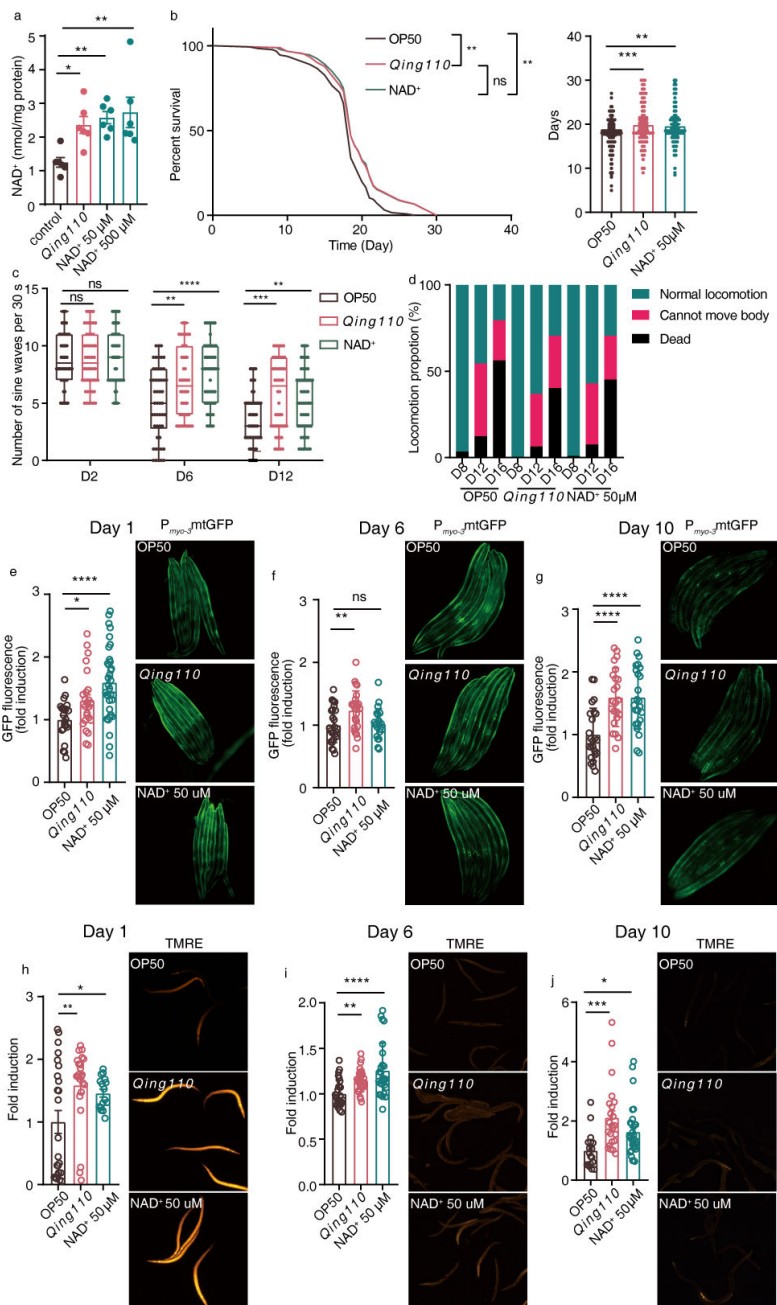

FIG 4 *Qing110* supplementation enhances lifespan, health span, and mitochondrial function in the *C. elegans* model. (a) Measurement of NAD$^+$ levels in day 3 adult N2 worms treated with *E. coli OP50*, *Qing110*, and NAD$^+$ at concentrations of 50 μM and 500 μM, respectively. (b) Survival curve and lifespan of N2 worms treated with *E. coli* OP50, *Qing110,* and NAD$^+$ starting at the L4 stage (*n* ≥ 90 worms for each group). (c) Crawling activity analysis by counting the number of sine waves (≥1 mm) that N2 worms crawled out in 30 s on days 2, 6, and 12. (d) Classification of locomotion ability in N2 worms on days 8, 12, and 16 (*n* > 20 worms for each group). (e) Images and relative GFP quantification of mitochondrial content in a muscle reporter strain (p$_{myo-3}$mito::GFP) on day 1 of adulthood. (f) Images and relative GFP quantification of mitochondrial content in p$_{myo-3}$mito::GFP on day 6. (g) Images and relative GFP quantification of mitochondrial content in p$_{myo-3}$mito::GFP on day 10. (h) Images and relative fluorescence quantification of mitochondrial membrane potential by staining with tetramethylr-hodamine ethyl ester (TMRE) dye in day 1 adults. (i) Images and relative fluorescence quantification of TMRE in day 6 adults. (j) Images and relative fluorescence quantification of TMRE in day 10 adults. Mean

**FIG 4** (Continued)

values are presented with error bars indicating SEM in panel (a). Means (±SEM) are plotted with each symbol representing an individual animal in panels (b, c, e-j). Statistical analysis was conducted using one-way ANOVA with Tukey's multiple comparisons test in panels (a, b, e-j), Log-rank (Mantel-Cox) test in panel (b), or two-way ANOVA with Tukey's multiple comparisons test in panel (c). "NS" indicates no significant difference ($P > 0.05$), while asterisks denote significance levels: $*P < 0.05$, $**P < 0.01$, $***P < 0.001$, $****P < 0.0001$. The data presented in panels (a, e-j) are representative of results from at least two independent experiments.

particularly of DNA double-strand breaks, underlies this condition (33). Past research has indicated that boosting NAD$^+$ levels in these mice can mitigate some symptoms and prolong both lifespan and health span (34, 35).

Our study aims to explore the possibility of using commensal bacteria as an alternative means of NAD$^+$ supplementation. We evaluated the NAD$^+$ content in the intestines of both WT and $Lyz1^{-/-}$ mice. Our findings revealed that $Lyz1^{-/-}$ mice exhibited elevated levels of NAD$^+$ level in the small intestinal epithelium compared to their WT counterparts (Fig. S6a). Furthermore, upon administering a high-fat diet, $Lyz1^{-/-}$ mice continued to demonstrate higher NAD$^+$ levels in the small intestinal epithelium relative to WT mice (Fig S6b). To evaluate the effect of $Qing110$ supplementation on NAD$^+$ content in mice, we administered $Qing110$ to WT mice for a period of 2 weeks and assessed the NAD$^+$ content across various tissues. The results showed an increase in NAD$^+$ levels and NAD$^+$/NADH ratio in the small intestinal epithelium, while NADH levels remained stable (Fig. 5a). However, the NAD$^+$ levels in the liver were unaffected by this 2-week $Qing110$ treatment (Fig S6c). We decided against prolonged $Qing110$ supplementation, due to the challenging nature of altering the homeostatic NAD$^+$ levels over the long term (36).

Next, we turned to examine the potential of $Qing110$ supplementation in prolonging the lifespan of $Atm^{-/-}$ mice. $Atm^{-/-}$ mice typically develop thymic lymphoma between 2 and 4 months of age, leading to their demise (37). We noted lymphomas in the enlarged thymus, weighing over 100 mg, in adult $Atm^{-/-}$ mice (Fig. S6d). By the age of 100 days, 10 out of 18 $Atm^{-/-}$ mice in the PBS control group had developed lymphoma, exhibiting thymus weights exceeding 100 mg. By contrast, only 5 out of 19 mice in the $Qing110$ treatment group had a thymus of such weight (Fig. 5b). Our subsequent survival analysis of the $Atm^{-/-}$ mice revealed a 66.3% increase in median lifespan (from 95 days to 158 days) and a 67.7% increase in maximum survival (from 195 days to 327 days) following $Qing110$ treatment (Fig. 5c).

Beyond lymphoma occurrence and reduced lifespan, $Atm^{-/-}$ mice also display a variety of abnormalities such as metabolic dysfunction, the diminished proportion of percentage B cells and T cells, and a loss of Purkinje cells in the cerebellum (34, 37–39). We further investigated whether $Qing110$ treatment could address these defects associated with ATM deficiency. $Qing110$ treatment significantly alleviated glucose intolerance in $Atm^{-/-}$ mice, as indicated by oral glucose tolerance tests (Fig. 5d and e), and enhanced insulin sensitivity compared to the PBS-treated $Atm^{-/-}$ mice, as shown by insulin tolerance tests (Fig. 5f and g). The treatment also appreciably increased the proportions of B cells and CD8$^+$ T cells while reducing myeloid cell proportions in peripheral blood (Fig. 5h; Fig. S6e). However, it did not impact the proportions of B cells and myeloid cells in the bone marrow (Fig. S6f-g). Further examination of Purkinje cell loss in $Atm^{-/-}$ mice showed a significant increase in cell numbers with $Qing110$ treatment compared to PBS treatment (Fig. 5i and j).

In conclusion, our findings underscore the considerable potential of $Qing110$ treatment in alleviating metabolic dysfunction, prolonging lifespan, and ameliorating other complications associated with ataxia telangiectasia in a mouse model.

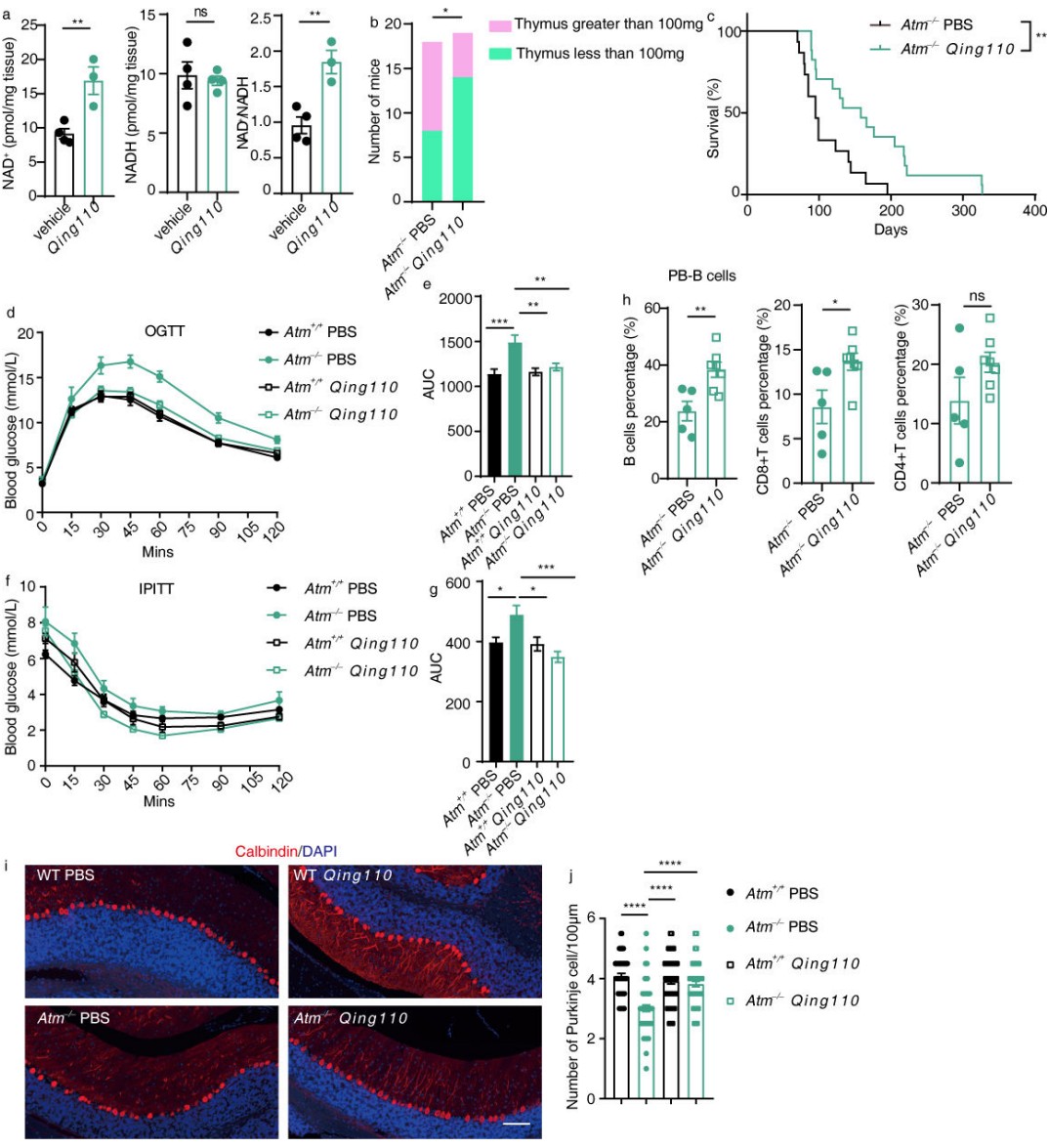

**FIG 5** *Qing110* supplementation exerts metabolic benefits and extends lifespan in the ataxia telangiectasia mouse model. (a) Measurement of NAD$^+$, NADH, and NAD(H) levels in small intestinal epithelial cells after treatment with *Qing110* and glycerol *via* oral gavage for 2 weeks. (b) The number of mice with thymus greater than 100 mg in 100-day-old *Atm*$^{-/-}$ mice treated with PBS or *Qing110* ($n$ = 18–19 mice/group). (c) Kaplan-Meier survival curves of *Atm*$^{-/-}$ mice, PBS- or *Qing110*-treated. *Atm*$^{-/-}$ mice were exposed to *Qing110* or PBS from 3 weeks of age and lifespan was determined ($n$ = 15–17 mice/group). (d) Plasma glucose profiles were measured during an oral glucose tolerance test (OGTT) in *Atm*$^{+/+}$ and *Atm*$^{-/-}$ mice, PBS- or *Qing110*-treated ($n$ = 11–12 mice/group). (e) Mean OGTT area under the curve measured in (d). (f) Plasma glucose profiles were measured during an insulin tolerance (ITT) in *Atm*$^{+/+}$ and *Atm*$^{-/-}$ mice, PBS- or *Qing110*-treated ($n$ = 5–9 mice/group). (g) Mean ITT area under the curve measured in (f). (h) The abundance of B-cell and T-cell subsets (CD4$^+$ and CD8$^+$) in peripheral blood was examined by flow cytometry and compared between *Atm*$^{-/-}$ mice treated with PBS or *Qing110*. (i) Confocal microscopy images of the Purkinje cells in the cerebellum of 3-month-old *Atm*$^{+/+}$ and *Atm*$^{-/-}$ mice, PBS- or *Qing110*-treated ($n$ = 3 ~ 5 mice/group). Scale bar = 100 μm. (j) Quantification of the numbers of Purkinje cells/100 μm ($n$ = 3–5 mice/group). Mean values are presented with error bars indicating SEM in panels (d-g, j). Means (±SEM) are plotted with each symbol representing an individual animal in panels (a, h). Statistical analysis was conducted using two-tailed Student's $t$-tests in panels (a, h), a one-sided chi-square test in panel (b), the log-rank (Mantel-Cox) test in panel (c), or one-way ANOVA with Tukey's multiple comparisons test in panels (e, g, j). "NS" indicates no significant difference ($P$ > 0.05), while asterisks denote significance levels: *$P$ < 0.05, **$P$ < 0.01, ***$P$ < 0.001, ****$P$ < 0.0001. The data presented in panels (d-g) are representative of results from at least two independent experiments.

## DISCUSSION

In this study, we sought to unravel the impact of intestinal lysozyme on host metabolism. Our findings indicated a substantial reduction in weight gain for mice lacking lysozyme, a trend that persisted in the face of an HFD challenge. The advantageous metabolic outcomes observed in $Lyz1^{-/-}$ mice were closely tied to changes in the gut microbiota. This was demonstrated by the elimination of these positive metabolic traits when $Lyz1^{-/-}$ mice cohabitated with WT mice, highlighting the crucial role of the microbiota in mediating these changes. Our in-depth analysis of the microbial composition in $Lyz1^{-/-}$ mice revealed a significant increase in the abundance of a particular OTU60. The bacterium from OTU60 was later identified as a novel species within the genus *ASTB*, designated as *ASTB Qinghuaensis Lab-110*. *ASTB Qing110* is particularly intriguing due to its ability to secrete $NAD^+$, a metabolite with known benefits across various organisms. To explore the potential of *Qing110* further, we conducted experiments using two different animal models. In *C. elegans*, supplementation with *ASTB Qing110* resulted in an extended lifespan and enhanced mitochondrial activity. Similarly, in $Atm^{-/-}$ mice, we observed marked improvements in metabolic dysfunction and an increase in lifespan with *ASTB Qing110* supplementation.

The manipulation of intestinal lysozyme appears to have complex implications for microbial communities and intestinal stability. Our findings add a new dimension to this complexity, demonstrating that lysozyme deficiency can protect against diet-induced obesity by modulating bacterial composition. However, this should be viewed in the broader context of lysozyme's effect on intestinal health, as it has been shown to both alleviate and exacerbate certain conditions depending on the circumstances (12, 40, 41).

Our comprehensive genomic analysis has firmly situated the isolated bacterium within the *ASTB* genus, under the *Erysipelotrichaceae* family. A notable observation is made when comparing the 16S rDNA sequences of *Qing110* to *M10-2*, a known species within the *ASTB* genus. They display a sequence identity of merely 91%, significantly below the standard threshold of 97% used for species delineation. Extensive genomic profiling of *Qing110* further confirmed its uniqueness, as it did not align with any known bacterial species.

Utilizing Average Nucleotide Identity (ANI) for a whole-genome analysis, we discovered a sequence identity of just 73% between *Qing110* and *M10-2*. This solidifies the argument that *Qing110* and *M10-2* are distinct species within the genus *ASTB*. It is crucial to highlight that the *ASTB* genus is still not completely characterized. Up until now, only three species have been described within this genus: *ASTB M10-2_s*, *EF603857_s*, and *FJ881277_s*, as recorded in the Ezbiocloud database (42). Furthermore, the Firmicutes phylum, to which ASTB belongs, has undergone significant taxonomic reorganizations, leading to the introduction of the term "*Erysipelatoclostridium*" (43). In light of the aforementioned data and the need for precise taxonomic classification, we propose that *Qing110* should be recognized and classified as a novel species within the *ASTB* genus of the *Erysipelotrichaceae* family (43).

$NAD^+$ is central to metabolism, serving as a co-enzyme in redox reactions and a crucial co-factor or substrate for $NAD^+$-dependent enzymes. It plays a critical role in a myriad of biological processes. Alterations in metabolic status, such as those induced by a high-fat diet, can lead to decreased $NAD^+$ levels, subsequently reducing the activity of $NAD^+$-dependent cellular processes. Counteracting this decrease, the supplementation of $NAD^+$ precursors like NR and NMN has been shown to protect against obesity induced by a high-fat diet in rodent models (44, 45).

Furthermore, studies have demonstrated that inhibiting $NAD^+$-consuming enzymes can also offer protection against high-fat diet-induced obesity. Mice with Parp1 or CD38 knockout, or those treated with PARP or CD38 inhibitors, exhibit elevated $NAD^+$ levels. These mice not only show resistance to obesity but also have enhanced metabolic rates during high-fat diets and aging (46–49). Nicotinamide phosphoribosyltransferase (NAMPT), a key enzyme in $NAD^+$ biosynthesis, is notably affected by high-fat diets, leading to reduced $NAD^+$ biosynthesis (45). Mice with a adipocyte-specific deletion of

NAMPT exhibit lower NAD$^+$ levels in their fat tissues, suffer from multi-organ insulin resistance and have impaired adipose tissue function. Remarkably, these issues can be ameliorated with NMN supplementation (50). These studies collectively provide compelling evidence that targeting NAD$^+$ metabolism can be an effective strategy against metabolic diseases.

NAD$^+$ metabolism involves intricate collaboration between microbes and their host. This collaboration has been observed through various mechanisms. Dietary supplementation of NAD$^+$ precursors, for instance, has been shown to elicit significant changes in the intestinal microbial community (51). Microbes themselves contribute to host NAD$^+$ metabolism through various mechanisms, including the deamination of dietary nicotinamide and nicotinamide riboside, which enhances the host NAD$^+$ metabolism (52). Recent research has further unveiled a symbiotic circuit of NAD$^+$ synthesis existing between the host and its microbiome (53). In addition, specific probiotic strains like *Akkermansia muciniphila* have been observed to offer therapeutic benefits in a murine model of Amyotrophic Lateral Sclerosis, potentially through the production of nicotinamide (54). This finding implies that augmenting host NAD$^+$ metabolism *via* probiotic intervention could serve as a viable strategy for ameliorating certain diseases. Collectively, these insights emphasize the essential role of microbial communities in modulating NAD$^+$ metabolism and underscore the potential of leveraging these host-microbe interactions for metabolic health optimization.

The intriguing capability of *Qing110* to secrete NAD$^+$ prompted us to delve deeper into its potential effects on host organisms. *Qing110* confers metabolic benefits on two different animal models. These results were consistent with the production of NAD$^+$ by *Qing110*. To further validate the pivotal role of NAD$^+$ in the beneficial effects of *Qing110* on host metabolism, forward genetics studies are warranted.

## MATERIALS AND METHODS

### Mice and housing conditions

*Lyz1$^{-/-}$* mice (C57BL/6J background) were described previously (24). *Atm$^{-/-}$* mice (129S6/SvEvTac background) were generously provided by Professor Baohua Liu (Shenzhen University Health Science Center) and were described previously (37). In this study, *Atm$^{-/-}$* and *Atm$^{+/+}$* mice were obtained as littermates from crosses of *Atm$^{+/-}$* +/−. WT C57BL/6J mice were sourced by the animal facility. 3-week-old WT 129S6/SvEvTac mice were procured from Vitalstar (Beijing, China) and acclimated in the animal facility. Throughout the study, mice of the same gender and age were matched, unless otherwise specified. All mice were housed in cages containing 3–6 animals each, maintained in a controlled environment with a 12/12-h light-dark cycle, and provided *ad libitum* access to food and water. The temperature was consistently regulated at 23 ± 2°C.

The animals were maintained on a standard chow diet, abbreviated as SFD (comprising 63.4% carbohydrates, 24.2% protein, and 12.4% fat). During high-fat diet treatment, mice that were 12 weeks old were transitioned to a high-fat diet, referred to as HFD (containing 60% of the kcal contained fat; D12492i, Research Diets Inc., New Brunswick, USA), and were provided *ad libitum* access for a duration of 8 weeks.

In the cohousing experiments, pairs of age- and gender-matched WT and *Lyz1$^{-/-}$* mice cohabited in cages in a 1:1 ratio after weaning. Meanwhile, separately housed mice were accommodated based on their respective genotypes.

### ASTB *Qing110* oral administration in mice

*ASTB Qing110* was cultivated in a modified GAM (Gifu Anaerobic Medium) with 0.1% L-cysteine, 0.1% resazurin, 2% vitamin K1, and 0.5% hemoglobin, all under anaerobic conditions at 37°C for a duration of 24 h. Following cultivation, the cultures were centrifuged at 4,500 g for 10 min, and the resulting supernatant was carefully removed. Subsequently, the pellet was resuspended in PBS containing 30% (vol/vol) glycerol.

$Atm^{-/-}$ mice and their age- and gender-matched $Atm^{+/+}$ littermates were treated with a suspension of $1 \times 10^9$ live *Qing110* bacteria in 0.2 mL PBS containing 30% glycerol *via* oral gavage. Control groups received an equivalent volume of the buffer *via* the same oral gavage method.

For the lifespan study, $Atm^{-/-}$ mice received daily oral administration of *Qing110* starting from the weaning period. Throughout the study, mice were subjected to daily monitoring, where any occurrences of mortality were meticulously recorded. In the event of a mouse's demise, necropsies were performed to gather relevant data. Lifespan curves were constructed employing the Kaplan-Meier method.

## Sample collection

Fecal samples were collected immediately upon defecation, rapidly frozen with liquid nitrogen, and subsequently stored at −80℃ for future analyses.

Various tissues, including the brain, subcutaneous adipose tissue (SAT), visceral adipose tissue (VAT), brown adipose tissue (BAT), liver, and cecal contents, were harvested. When necessary, these tissues were weighed, dissected, promptly immersed in liquid nitrogen to preserve their integrity, and then stored at −80℃ for future biochemical analysis.

To isolate intestinal epithelial cells, the following procedures were followed (55). After the removal of fat and connective tissues, the small intestine was longitudinally opened and rinsed with cold PBS. The tissue was then vigorously washed in 20 mL cold PBS for 1 min and rinsed once more in cold PBS. The intestine tissues were subjected to digestion in 20 mL PBS containing 1.5 mM EDTA and 0.5 mM DTT at 4℃ for 15 min. Following this, the tissues were transferred into cold PBS. Epithelial cells were dissociated by vigorous vortexing for 1 min in cold PBS. Supernatants were collected and the intestinal tissues were transferred to fresh PBS with 1.5 mM EDTA and 0.5 mM DTT, followed by an additional 15 min at 4℃. The vortexing procedure was repeated twice. Supernatants from all washes were combined and subjected to centrifugation to pellet the epithelial cells. Isolated cells were stored at −80℃ for subsequent quantification of $NAD^+$.

## Oil red O staining

Tissues (liver and BAT) were collected from euthanized mice and fixed in 4% PFA at 4℃ overnight and then incubated in 30% sucrose for 12 h at 4℃. Fixed tissues were embedded in the Tissue-Tek O.C.T. Compound (Sakura Finetek, Torrance, CA, USA). Serial sections (12 µm in thickness) were made and stained with oil red O (BBI Life Science) working solution (0.5% oil red O in isopropanol:ddH$_2$O = 3:2, vol/vol) for 10 min, counter-stained with hematoxylin (Sigma-Aldrich). Images were captured using the 3DHISTECH SCAN II scanner (3DHISTECH, Budapest, Hungary).

## Liver triglyceride assay

Livers were carefully harvested from euthanized mice. Approximately 50 mg of liver tissue was homogenized in 500 µL of cold dichloromethane/methanol (2:1, vol/vol). Subsequently, 125 µL of ddH$_2$O was added to the tissue lysate. The resulting mixture was subjected to centrifugation at 12,000 rpm for 15 min. After centrifugation, 100 µL of the organic fraction was carefully transferred to a fresh tube. This fraction was then dried under a stream of nitrogen. The resulting pellets were subsequently reconstituted in 200 µL of ethanol. The concentration of triglycerides was determined using a commercially available TG kit (BIOSINO, China).

## Histological analyses

Tissue samples, including SAT, VAT, BAT, liver, and cerebellum, were initially fixed in 4% paraformaldehyde (PFA) at 4℃ overnight. After fixation, the samples were dehydrated in a gradient ethanol solution and subsequently embedded in paraffin. 6-µm-thick slices

were prepared from the paraffin block. The slices were subjected to hematoxylin and eosin (H&E) staining. The stained slices were visualized and captured using a 3DHISTECH SCAN II scanner (3DHISTECH, Budapest, Hungary). For the quantification of adipocyte size and distribution in VAT and SAT, 3–5 fields per sample were analyzed using Image J software. In brown adipose tissue, lipid droplets were identified by white areas and quantified from 3 to 5 fields per sample using Image J software. To assess inflammation, we counted crown-like structures (CLSs) in SAT and VAT, which serve as an indicator of inflammation. This counting was done in 3–5 fields per sample at 10× magnification.

## Purkinje cell immunofluorescence staining and quantification

6-µm-thick mouse cerebellar paraffin slices were utilized for immunofluorescence staining. The staining procedure followed a previously described method (24). Tissue slices were initially mounted on positively charged glass slides and subjected to dewaxing. To retrieve antigens, the slices were immersed in 0.01 M sodium citrate buffer (pH = 6.0) and streamed for 25 min. Slides were blocked with a normal goat serum blocking reagent (Wuhan Boster, China) for 30 min to prevent non-specific binding. Subsequently, the slices were incubated with an anti-Calbindin D-28K antibody (Sigma) at 4°C overnight to label the Purkinje cells. After primary antibody incubation, the slices were exposed to a fluorophore-conjugated secondary antibody at room temperature for 2 h. The slices were counterstained with DAPI (4′,6-diamidino-2-phenylindole) and mounted in Fluoromount-G (SouthernBiotech). Confocal images were captured with an Axio Scan. Z1 Zeiss imaging system with a 10× objective. The quantification of Purkinje cells was performed by counting the number of Purkinje cells per 100 µm (56). For quantification purposes, 10–15 images were assessed for each mouse, with 3–5 mice per group included in each experimental group. This method allowed for the evaluation of Purkinje cell numbers in a systemic and comprehensive manner.

## Body composition analysis

The body composition of animals was assessed using an EchoMRI body composition analyzer system (EchoMRI 100 system, EchoMRI Medical System, Houston, TX). Briefly, conscious mice were first weighed to record their total body weight. Each mouse was then placed in a thin-walled plastic cylinder specifically designed for this purpose. Within the cylinder, a cylindrical plastic insert was added to restrict the movement of mice during the analysis. The restrained mice were briefly exposed to a low-intensity electromagnetic field generated by the EchoMRI system. During the exposure, the system measured and quantified the fat mass and lean mass.

## Oral glucose tolerance test (OGTT) and intraperitoneal glucose tolerance test (ipGTT)

For the glucose tolerance tests, mice were subjected to a 16-h fasting period. In the OGTT, glucose (2 grams per kilogram of body weight) was administered orally as a 20% glucose solution *via* gavage. Blood glucose was measured at various time points: before glucose administration (0 min) and 15, 30, 45, 60, 90, and 120 min after glucose administration. In the ipGTT, glucose (2 grams per kilogram of body weight) was injected into the peritoneal cavity as a 20% glucose solution. Blood glucose levels were monitored at the same time intervals as in the OGTT.

Blood samples were obtained from the tail tip and analyzed using glucose test strips (Abbott FreeStyle Optium) and a blood glucometer (Abbott FreeStyle Optium Neo).

## Insulin tolerance test

Mice, which had fasted for 6 h prior to the test, received an intraperitoneal injection of insulin at a dose of 0.75 IU/kg. Blood glucose levels were measured at various time

points: before insulin administration (0 min) and 15, 30, 45, 60, 90, and 120 min after insulin administration. Blood samples were obtained from the tail tip and analyzed using glucose test strips (Abbott FreeStyle Optium) and a blood glucometer (Abbott FreeStyle Optium Neo).

## Metabolic cages

Mice were housed individually for a 2-day acclimatization period before metabolic analysis. Indirect calorimetric measurements, including oxygen ($O_2$) consumption, carbon dioxide ($CO_2$) expiration, energy expenditure (EE), RER, and food consumption were recorded over a 4-day period using the Phenomaster system (TSE Systems GmbH, Bad Homburg, Germany). Data obtained from the metabolic studies were analyzed and graphed using GraphPad Prism.

## Gene expression analysis using quantitative reverse-transcription PCR (qPCR)

RNA was extracted from various tissue samples using Trizol. Subsequently, the exacted RNA was reversely transcribed into cDNA using the PrimerScript RT reagent kit (Takara, Japan). For quantitative PCR (qPCR), triplicate reactions were carried out using SYBR Premix Taq (Takara, Japan) on an ABI QuantStudio 6 thermal cycler. The qPCR thermal cycling conditions comprised an initial denaturation step at 95℃ for 30 seconds, followed by 40 cycles of denaturation at 95℃ for 5 seconds and annealing/extension at 60℃ for 34 seconds. To confirm the specificity of qPCR, melting curves were generated for each PCR reaction. The relative levels of target mRNA were determined by calculating Delta Delta Ct values between the target gene and the control. Details of the primer sequences used for qPCR can be found in the supplementary information.

## Microbial community analysis of fecal samples

Total DNA was extracted from fecal samples with a QIAamp fast DNA stool mini kit (Qiagen, Valencia, CA, USA) following the manufacturer's handbook. After assessing the DNA quantity and quality, an amplicon sequencing library was constructed. The library was based on the PCR-amplified V3-V4 variable regions of the 16S rRNA gene. The library was paired-end sequencing on an Illumina MiSeq platform, following the manufacturer's procedures. The Illumina sequencing process generated raw fastq files. These sequence reads underwent merging, trimming, filtering, alignment, and clustering. The analysis utilized UPARSE (version 7.0.1090, http://drive5.com/uparse/) for the above steps. The taxonomic classification of each 16S rRNA gene sequence was carried out using the RDP Classifier algorithm (version 2.11, https://sourceforge.net/projects/rdp-classifier/) with a confidence threshold of 70%. The silva138 16S rRNA database (http://www.arb-silva.de) was employed for this classification. Operational taxonomic unit (OTU) sequences were clustered, with those having over 97% similarity grouped into the same OTU. Alpha diversity metrics at the OTU level, based on the Shanon index, Simpson index, Ace index, and Chao index, were calculated by the MOTHUR program (version 1.30.0). Beta diversity at the OTU level, using the Bray-Curtis method, was qualitatively examined by the MOTHUR program and visualized through Principal Component Analysis (PCA) using the vegan R packages.

For visualization purposes, the bacterial taxonomic distributions of each group community were displayed at the genus level using the vegan R packages. Additional analyses of alpha diversity, beta diversity, bacterial taxonomic distributions, and heatmap generation were performed at the Majorbio Cloud platform.

The 16S rRNA gene sequencing data have been submitted to NCBI Sequence Read Archive (SRA) with the accession number PRJNA898979.

## Bacterial isolation and identification

Fecal samples from mice were resuspended in 1 mL of PBS containing 0.1% L-cysteine hydrochloride. These samples were then serially diluted to generate samples with

varying bacterial concentrations. The diluted samples were plated on Gifu anaerobic broth blood agar plates, and cultured under aerobic conditions at 37°C for 24 to 72 h. Single colonies that formed on the plates were selected and transferred into a specialized anaerobic cell for further culturing. Genomic DNA was extracted from the cultured bacteria within the anaerobic cell using the TIANamp Genomic DNA Kit (Tiangen, China). The full-length 16S rRNA gene of these bacteria was amplified with specific barcoded primers. Sanger sequencing was performed on the PCR products to obtain the full-length 16S rRNA gene sequences. The obtained sequences were aligned and taxonomically assigned using the Ezbiocloud data set. Bacteria that were among the top 20 in terms of abundance were selected for further analysis. Frozen stocks of these isolated bacterial strains were prepared and stored at −80°C for future use.

## Taxonomic classification of *Qing110*

After identifying the bacterial strain in OTU60, whole-genome sequencing was performed using a combination of Pacbio and Illumina PE150 technologies. A complete genome was assembled. To determine the genetic relatedness of *Qing110* to other bacteria, the ANI of its genome was calculated. This analysis was carried out in comparison to the genomes of bacteria that were taxonomically close to *Qing110*. The ANI analysis revealed that *Qing110* was most closely related to the following taxonomic classification: Bacteria, Firmicutes, Erysipelotrichia, Erysipelotrichales, *Erysipelotrichaceae,* and *ASTB_g*. Based on this analysis, *Qing110* was classified into the *ASTB* genus, and its species name was redefined as "*ASTB Qinghuaensis*." The strain designation was specified as "Lab-110."

## *Qing110* quantification by qPCR

Genomic DNA was extracted from mouse feces using the QIAamp DNA Stool Mini Kit (Tiangen, China). The quantity and quality of DNA were assessed using a NanoDrop2000 (Thermo Fisher Scientific, USA). *Qing110*-specific primers were designed based on the 16S rRNA gene sequences of the *Qing110* strain, which were obtained from whole-genome sequencing. The designed primers for *Qing110* were as follows:

*Qing110* Forward Primer: 5′-CATGCAAGTCGAACGAGGGTC-3′

*Qing110* Reverse Primer: 5′-CGATGCCGTCTCTGTCCCTA-3′. In addition, primers for total bacteria were used for comparison:

Total Bacteria Forward Primer: 5′-ACTCCTACGGGAGGCAGCAGT-3′

Total Bacteria Reverse Primer: 5′-ATTACCGCGGCTGCTGGC-3′.

The qPCRs were performed with the designed primers. The reaction conditions included an initial denaturation step at 95°C for 30 seconds, followed by 40 cycles of denaturation at 95°C for 5 seconds and annealing/extension at 60°C for 34 seconds. To determine the relative abundance of *Qing110*, the obtained qPCR data were normalized to the total bacterial load. This normalization step allowed for the quantification of *Qing110* relative to the overall bacterial population in the fecal samples.

## *C. elegans strains and synchronization*

Strains used were wild-type Bristol N2, and SJ4103 (*zcIs14[myo-3::GFP(mit)]*). Strains were from the Caenorhabditis Genetics Center (University of Minnesota). Worms were grown on a standard nematode growth medium (NGM) at 20°C, following the standard protocols as described previously (57).

To assess the effect of *Qing110* on *C. elegans,* the following synchronization procedure was employed. Gravid adult worms were subjected to a bleach treatment using a mixed solution of NaClO and NaOH for 90 seconds. The bleached worms were washed three times with M9 buffer. Eggs obtained from the bleach treatment were placed on NGM plates without *E. coli* OP50 and left for 12 h to allow them to develop into L1 worms. L1 worms were then transferred to NGM plates with *E. coli* OP50 to generate a synchronized

population of L4 worms. L4 worms were subsequently transferred to new plates and treated with *E. coli* OP50, *Qing110* or NAD+.

To prepare *Qing110*-containing plates, *Qing110* and *E. coli* OP50 were separately cultured, pelleted, and washed. *E. coli* OP50 and *Qing110* were mixed by a 1:2 ratio and resuspended in M9 buffer. The mixture was then plated onto NGM plates, ensuring complete coverage to prevent worms from avoiding the bacteria. For NAD+ supplementation, NAD+ was added at the indicated concentration just before pouring the plates. Worms were transferred to new plates every other day during the experiment.

### *C. elegans* development and lifespan assay

Synchronized L4 worms were distributed evenly onto new plates with (1) *E. coli* OP50 (2), *E. coli* OP50, and *Qing110* in a 1:2 ratio, or (3) *E. coli* OP50 supplemented with 50 µM NAD+. On Day 2 and day 3, adult worms from each group were randomly chosen for body length measurement using a Nikon TI1 CCD camera. Each experiment included more than 18 worms for each group per experiment.

Worm lifespan tests were carried out as described (58). For each experimental condition, more than 90 worms were used. Worms were scored every other day throughout the lifespan assay.

### *C. elegans* brood size assay

Worms were synchronized and cultured as described earlier. Day 0 adult worms were transferred to new plates daily. The number of eggs were laid on each plate was counted every 24 h until no more eggs laid (59). All plates were then cultured for an additional 2–3 days until reaching the L3–L4 period. The number of offspring produced by each worm was counted.

### *C. elegans* locomotion assay

Worms were synchronized and cultured as described earlier. On days 2, 6, and 12, a minimum of 20 worms were randomly selected for locomotion assessment using two indicators: movement speed and movement trajectory. The number of sine waves (≥1 mm) in 30 seconds was counted, as described (60). The movement trajectory of worms at different time points was assessed using a scoring method as described (60).

### *C. elegans* heat stress resistance assay

Worms were synchronized and cultured as described. On day 2 and day 8, adult worms from each group were transferred from 20°C to 37°C and their response was assessed every hour. The number of dead worms was counted, and the mean survival time was calculated.

### Mitochondrial mass assay

Mitochondrial mass was measured as previously described (61). Synchronized *zcls14 [myo-3::GFP(mit)]* worms were grown until the L4 larval stage and then fed on indicated plates. On Day 1, day 6, and day 10, adults *C. elegans* hermaphrodites were anesthetized with 50 mM levamisole. They were mounted on 3% agarose pads and maintained at room temperature. Photos were taken under blue excitation light (488 nm) using the inverted Nikon Eclipse Ti2 microscope (Nikon Ti2-E). Fluorescence was quantified with Image J software (http://rsbweb.nih.gov/ij/). Three independent experiments were conducted with at least 20 worms per experiment.

### TMRE assay

The TMRE staining experiment followed a previously established method (61). Synchronized N2 *C. elegans* were grown until the L4 larval stage and then fed on indicated plates. Day 1, day 6, and day 10 adult worms were transferred to NGM agar plates containing

30 µM TMRE dissolved with heat-inactivated *E. coli* OP50 or mixture of *E. coli* OP50 and *Qing110* (30 min, 65℃). Worms were left on the plates for 2 h. The worms were subsequently washed with M9 and transferred to fresh NGM plates with heat-inactivated bacteria for 1 h. The animals were washed and prepared for imaging as described in the mitochondrial mass assay. Fluorescence was quantified with Image J software. Two independent experiments were conducted with at least 15 worms per experiment.

## NAD$^+$ measurement

The NAD$^+$ levels in day 3 adult worms were measured using a commercial NAD$^+$/NADH assay kit (Beyotime, China). Worms were synchronized and cultured. Day 3 adult worms were homogenized in NAD$^+$/NADH extraction solution. The worm homogenate was centrifuged at $12,000 \times g$ for 10 min at 4℃, and the supernatant was collected. In a 96-well plate, 20 µL of the supernatant was mixed with 90 µL of ethanol dehydrogenase working solution. The mixture was incubated at 37℃ in the dark for 10 min. Subsequently, 10 µL of the chromogenic solution was added to each reaction, mixed, and incubated in the dark at 37℃ for 30 min. Measurements at the absorbance at 450 nm were taken to determine the total amount of NAD$^+$ and NADH. For NADH measurement, 100 µL of the supernatant was heated for 30 min at 60℃ to decompose NAD$^+$. Then, 20 µL of the heat-treated supernatant was taken to determine the content of NADH. The content of NAD$^+$ was calculated by subtracting the amount of NADH from the total amount of NAD$^+$ and NADH.

To measure the amount of NAD$^+$ in mouse tissue, 30 mg of tissues was homogenized in 900 µL 80% methanol. The homogenate was placed at −80℃ for 1 h. After the incubation, the homogenate was centrifuged at 12,000 rpm for 10 min at 4℃. The supernatant was collected. NAD$^+$ levels in the supernatant were quantified using liquid chromatography–mass spectrometry (LC-MS).

## Statistical analyses

The data are presented as means ± SEM. The normality of the data was assessed using the Kolmogorov-Smirnov test. For most of the data, statistical significance between groups was determined using one-way ANOVA with Tukey's multiple comparisons test post hoc test and two-tailed Student's *t*-tests. Alpha diversity and principal coordinates in the 16S rRNA sequencing analysis were evaluated using the Wilcoxon rank-sum test. Microbial community clustering was assessed using Analysis of Similarities (ANOSIM) based on unweighted UniFrac distance matrices. *P*-values less than 0.05 were considered statistically significant. All statistical analyses were performed using GraphPad Prism.

## ACKNOWLEDGMENTS

The authors would like to thank Professor Baohua Liu from Shenzhen University Health Science Center and Professor Guangshuo Ou from the School of Life Sciences, Tsinghua University for sharing animals and technique help. The authors would like to thank the Facility Center of Metabolomics for their technical assistance and support.

Z.L. conceived and directed the research. C.Z., C.X., and K.Z. designed, performed, and analyzed the majority of experiments. X.L., G.Q., Y.Z., K.S., and K.D. performed the experiments. C.Z. and C.X. prepared the manuscript. C.Z and K.Z. revised the manuscript. All authors read the manuscript, provided feedback, and approved the final manuscript.

This study was supported by the National Key Research and Development Program of China (Grant no. 2017YFA0503403 and 2018YFE0207300) and the National Natural Science Foundation of China (Grant no. 31730028, to Z.L.).

The authors have stated that there are no conflicts of interest to disclose.

## AUTHOR AFFILIATIONS

[1]Institute for Immunology, School of Medicine, Tsinghua University, Beijing, China

[2]Beijing Institute of Genomics, Chinese Academy of Sciences, China National Center for Bioinformation, Beijing, China
[3]University of Chinese Academy of Sciences, Beijing, China
[4]Tsinghua-Peking Center for Life Sciences, Tsinghua University, Beijing, China

## AUTHOR ORCIDs

Chengye Zhang ⓘ http://orcid.org/0009-0004-3628-7936
Zhihua Liu ⓘ http://orcid.org/0000-0002-0269-0901

## FUNDING

| Funder | Grant(s) | Author(s) |
| --- | --- | --- |
| MOST \| National Key Research and Development Program of China (NKPs) | 2017YFA0503403, 2018YFE0207300 | Zhihua Liu |
| MOST \| National Natural Science Foundation of China (NSFC) | 31730028 | Zhihua Liu |

## AUTHOR CONTRIBUTIONS

Chengye Zhang, Conceptualization, Data curation, Formal analysis, Investigation, Methodology, Project administration, Writing – original draft, Writing – review and editing | Chen Xiang, Conceptualization, Data curation, Investigation, Methodology, Project administration, Writing – original draft | Kaichen Zhou, Data curation, Methodology, Software, Writing – review and editing.

## DATA AVAILABILITY

Data are available in a public, open-access repository. All data relevant to the study are included in the article or uploaded as supplementary information. All datasets and raw data generated and analyzed during the current study are available from the corresponding author upon request. The 16S rRNA gene sequencing raw sequences of the mouse study can be accessed in the database with an accession code.

## ETHICS APPROVAL

The animal experiments described in the Methods section were conducted in strict accordance with the guidelines and protocols established by Tsinghua University (protocol number 21-LZH-2). All procedures involving animals were approved by the Institutional Animal Care and Use Committee (IACUC) of Tsinghua University. The animal experiments were conducted in compliance with the principles and standards set forth by the Association for Assessment and Accreditation of Laboratory Animal Care (AAALAC). To minimize suffering, mice were humanely sacrificed using hypoxia via carbon dioxide inhalation, followed by cervical dislocation, immediately upon completion of the intended experiments.

## ADDITIONAL FILES

The following material is available online.

### Supplemental Material

**Supplemental Material (mSystems01214-23-s0001.docx).** Supplemental figures.

### Open Peer Review

**PEER REVIEW HISTORY (review-history.pdf).** An accounting of the reviewer comments and feedback.

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
