## [Reviewer comments · mSystems]

Intestinal Lysozyme1 Deficiency Alters Microbiota Composition and Impacts Host Metabolism Through the Emergence of NAD⁺-Secreting ASTB Qing110 Bacteria

Chengye Zhang, Chen Xiang, Kaichen Zhou, Xingchen Liu, Guofeng Qiao, Yabo Zhao, Kemeng Dong, Ke Sun, and Zihua Liu

Corresponding Author(s): Zihua Liu, Tsinghua University

Review Timeline:

Submission Date:	November 14, 2023
Editorial Decision:	December 20, 2023
Revision Received:	January 20, 2024
Accepted:	January 26, 2024

Editor: Hongwei Zhou

Reviewer(s): The reviewers have opted to remain anonymous.

Transaction Report:

DOI: <https://doi.org/10.1128/msystems.01214-23>

Re: mSystems01214-23 (**Intestinal Lysozyme Deficiency Alters Microbiota Composition and Impacts Host Metabolism Through the Emergence of NAD⁺-Secreting ASTB Qing110 Bacteria**)

Dear Prof. Zhihua Liu:

Revision Guidelines

Sincerely,
Hongwei Zhou
Editor
mSystems

Reviewer #1 (Comments for the Author):

The manuscript by Zhang et al described the microbiota features associated with the metabolic alterations in lysozyme deficiency and identified a previously unknown gut bacteria strain, ASTB Qing110, enriched in the microbiota of lysozyme-deficient mice. The authors also characterized the genomic and metabolic features of ASTB Qing110 and found that ASTB Qing110 was capable of producing NAD⁺. The authors then showed evidence that ASTB Qing110 could ameliorate disease

conditions correlating with NAD⁺ production by the bacteria. The work identifies a novel metabolically-important gut microbe and provides key insights into its physiological function. However, the reviewer thinks there are a few questions to be addressed.

Major concerns:

- 1) Do NAD⁺ levels differ in the intestine or feces between WT and Lyz1^{-/-} mice?
- 2) How does the abundance of ASTB Qing110 change in response to HFD feeding?
- 3) In mice and worms treated with ASTB Qing110, what is the abundance of ASTB Qing110 achieved by the treatment? Does ASTB Qing110 treatment impact other components of the gut microbiota?

Minor concern:

- 1) As the Lyz1^{-/-} mouse model used is a whole-body knockout, it is not appropriate to state "Intestinal Lysozyme Deficiency" in the title or the author should provide evidence that other body parts are not impacted.
- 2) The authors start with characterizing metabolism and HFD-induced obesity in Lyz1^{-/-} mice and identify ASTB Qing110 as a significantly enriched gut microbe. I would suggest including the role of NAD⁺ in obesity and metabolic disorders in the discussion.

Response Letter

We'd like to express our gratitude to the reviewer for their comments and suggestions. We have carried out substantial new experimentation addressing all of the questions and concerns. These additional data have strengthened and clarified the revised manuscript. We have also compiled a point-by-point response to the reviewer. The corresponding changes are in yellow in main text.

Comments for the Author: The manuscript by Zhang et al described the microbiota features associated with the metabolic alterations in lysozyme deficiency and identified a previously unknown gut bacteria strain, ASTB Qing110, enriched in the microbiota of lysozyme-deficient mice. The authors also characterized the genomic and metabolic features of ASTB Qing110 and found that ASTB Qing110 was capable of producing NAD⁺. The authors then showed evidence that ASTB Qing110 could ameliorate disease conditions correlating with NAD⁺ production by the bacteria. The work identifies a novel metabolically-important gut microbe and provides key insights into its physiological function. However, the reviewer thinks there are a few questions to be addressed.

Response: We thank the Reviewer's kind comments on our manuscript and appreciate the helpful suggestions.

Major concerns:

1) Do NAD⁺ levels differ in the intestine or feces between WT and *Lyz1*^{-/-} mice?

Response: We express our gratitude to the reviewer for their insightful comment. In response, we conducted an evaluation of the NAD⁺ content in the intestines of both WT and *Lyz1*^{-/-} mice. Our findings revealed that *Lyz1*^{-/-} mice exhibited elevated levels of NAD⁺ level in the small intestinal epithelium compared to their WT counterparts, as depicted in RL-Fig.1a. Furthermore, upon administering a high-fat diet, *Lyz1*^{-/-} mice continued to demonstrate higher NAD⁺ levels in the small intestinal epithelium relative to WT mice, as shown in RL-Fig.1b. This additional data has been duly incorporated into our revised manuscript, specifically in Fig S6.

RL-Fig.1 NAD⁺ levels in the small intestinal epithelium in WT and *Lyz1*^{-/-} mice on SFD(a) and HFD(b). The NAD⁺ content was quantified using liquid chromatography–mass spectrometry (LC-MS). Mean and s.e.m. are plotted, with each dot representing one individual animal.

2) How does the abundance of *ASTB Qing110* change in response to HFD feeding?

Response: We appreciate the Reviewer for the comment. *Qing110* plays a pivotal role in the metabolic advantages seen in *Lyz1^{-/-}* mice. We assessed the abundance of *ASTB Qing110* in *Lyz1^{-/-}* mice during a high-fat diet regimen. Our findings reveal a significant increase in *Qing110* levels on day 14, with no notable changes on days 7 and 21. This suggests that *Qing110*'s abundance remains stable throughout high-fat feeding, as depicted in RL-Fig.2a. Further exploring the impact of a high-fat diet (HFD) on *Qing110* colonization, we orally administered *Qing110* to wild-type (WT) mice on an HFD. The results showed that the average abundance of *Qing110* was 4.065% (RL-Fig.2b), indicating that HFD does not impede *Qing110* colonization. These results have been incorporated into our revised manuscript (see Fig 3).

RL-Fig.2 The abundance of *ASTB Qing110* in response to HFD feeding. (a) Real-time PCR analysis of *Qing110* in fecal samples from *Lyz1^{-/-}* mice on SFD and HFD. The relative abundance of *Qing110* was quantified by normalizing to total bacteria, and the relative abundance of *Qing110* on HFD was normalized to the relative abundance of *Qing110* on SFD. (b) Real-time PCR analysis of *Qing110* in fecal samples from WT mice treated with PBS or *Qing110* for 1 month. The relative abundance of *Qing110* was quantified by normalizing to total bacteria. Mean and s.e.m. are plotted, with each dot representing one individual animal.

3) In mice and worms treated with *ASTB Qing110*, what is the abundance of *ASTB Qing110* achieved by the treatment? Does *ASTB Qing110* treatment impact other components of the gut microbiota?

Response: We are grateful for the Reviewer's feedback. We assessed the relative abundance of *Qing110* in WT mice treated with *Qing110*. The results showed the average relative abundance of *Qing110* in *Qing110*-treated mice was 1.8% (RL-Fig.3a). To explore *Qing110*'s impact on the gut microbiota's other components, we initially administered antibiotics (ABX) orally to WT mice for one week. This was followed by a

three-month daily oral administration of *Qing110*. Post-treatment, we analyzed the microbial communities in both vehicle and *Qing110*-treated mice using 16S ribosomal RNA gene sequencing. A Principal Component Analysis (PCA) based on Bray-Curtis distances highlighted distinct taxonomic composition differences between the control and *Qing110*-treated mice (RL-Fig.3b). In a detailed comparison of the relative abundance, significant differences were identified for *Erysipelotrichaceae* which family *Qing110* belongs to. Other families with lower abundance also have significant differences, such as *UCG-010*, *Atopobiaceae*, *Peptococcaceae*, *Saccharimonadaceae* (RL-Fig.3c-d). Overall, oral administration of *Qing110* significantly altered the composition of gut microbiota in mice, with *Erysipelotrichaceae* family being the most significantly increased. We have included the information in our revised manuscript (Fig S3).

RL-Fig.3 The effect of oral administration of *Qing110* on the gut microbiota of mice. (a) Real-time PCR analysis of *Qing110* in fecal samples from vehicle or *Qing110* treated mice. The relative abundance of *Qing110* was quantified by normalizing to total bacteria. (b) PCA analysis of fecal microbiota based on the relative abundance of bacterial OTUs. (c) Relative abundance on family level in the feces of control and *Qing110*-treated WT mice. (d) Relative abundance and statistical description of different families. Mean and s.e.m. are plotted, with each dot representing one individual animal.

Minor concern:

1) As the *Lyz1*^{-/-} mouse model used is a whole-body knockout, it is not appropriate to state "Intestinal Lysozyme Deficiency" in the title or the author should provide evidence that other body parts are not impacted.

Response: We are thankful for the Reviewer's comment. In the mouse genome, there are two lysozyme genes: lysozyme M (lysozyme 2) and lysozyme P (lysozyme 1). Lysozyme M is expressed in myeloid cells, while *Lyz1*, also known as lysozyme P, is expressed in intestinal Paneth cells (Cross M et al, Proc Natl Acad Sci U S A, 1988; Klockars M and EF., J Histochem Cytochem, 1974). Paneth cells, located at the base of intestinal crypts, secrete a significant amount of *Lyz1* into the intestinal lumen. *Lyz1*, being the primary lysozyme in the gut lumen, is also known as intestinal lysozyme.

Our laboratory created *Lyz1* knockout mice and verified lysozyme expression through various methods. Immunohistochemical and immunofluorescence staining on ileum paraffin sections from *Lyz1*^{-/-} mice showed an absence of lysozyme in their Paneth cells (RL-Fig.4A and RL-Fig.4B). Immunoblotting experiments further confirmed the lack of lysozyme expression in the crypts of *Lyz1*^{-/-} mice (RL-Fig.4C). Additionally, we conducted whole tissue immunofluorescence staining of the small intestine, using antibodies to label lysozyme and defensin, and FITC-labeled lectin to label mucin. In WT mice, lysozyme colocalized with mucin, but was absent in *Lyz1*^{-/-} mice (RL-Fig.4D). Procryptdin, however, colocalized with mucin in both WT and *Lyz1*^{-/-} mice (RL-Fig.4E), indicating that lysozyme absence does not impact Paneth cell exocytosis. Furthermore, *Lyz2* mRNA levels in myeloid and Paneth cells of *Lyz1*^{-/-} and WT mice were not significantly different (RL-Fig.5) (Zhang et al., Cell Res, 2019).

Following the Reviewer's suggestion, we have revised the manuscript title to "Intestinal Lysozyme1 Deficiency" for greater clarity and to prevent any potential misunderstandings.

RL-Fig.4 *Lyz1*^{-/-} mice lack lysozyme in Paneth cells and intestinal lumen. (A) IHC staining of lysozyme in Paneth cells of the ileum from WT and *Lyz1*^{-/-} mice. (B) Immunostaining and confocal imaging of lysozyme in paraffin sections of ileum from WT and *Lyz1*^{-/-} mice. Green represents lysozyme, and blue represents cell nucleus. Scale bar=10 μm. (C) Immunoblotting analysis of lysozyme in the crypts of WT and *Lyz1*^{-/-} mice. (D-E) Immunostaining and confocal imaging of lysozyme, prosecretin and mucin in paraffin sections of small intestine from WT and *Lyz1*^{-/-} mice. Red represents (D) lysozyme or (E) prosecretin, while green represents mucin. Scale bar=20 μm.

RL-Fig.5 The relative levels of *Lyz1* (a) and *Lyz2* (b) mRNA in Paneth cells and bone marrow cells (Zhang et al., *Cell Res*, 2019).

2) The authors start with characterizing metabolism and HFD-induced obesity in *Lyz1*^{-/-} mice and identify *ASTB Qing110* as a significantly enriched gut microbe. I would suggest including the role of NAD⁺ in obesity and metabolic disorders in the discussion.

Response: We are grateful for the Reviewer's suggestion to enrich the discussion in our article. NAD⁺ is central to metabolism, serving as a co-enzyme in redox reactions and a crucial co-factor or substrate for NAD⁺-dependent enzymes. It plays a critical role in a myriad of biological processes. Alterations in metabolic status, such as those induced by a high-fat diet, can lead to decreased NAD⁺ levels, subsequently reducing the activity of NAD⁺-dependent cellular processes. Counteracting this decrease, the supplementation of NAD⁺ precursors like NR and NMN has been shown to protect against obesity induced by a high-fat diet in rodent models (Canto et al., *Cell Metab*, 2012; Yoshino et al., *Cell Metab*, 2011).

Furthermore, studies have demonstrated that inhibiting NAD⁺ consuming enzymes can also offer protection against high-fat diet-induced obesity. Mice with *Parp1* or *Cd38* knockout, or those treated with PARP or CD38 inhibitors, exhibit elevated NAD⁺ levels. These mice not only show resistance to obesity but also have enhanced metabolic rates during high-fat diets and aging (Bai et al., *Cell Metab*, 2011; Barbosa et al., *FASEB J*, 2007; Camacho-Pereira et al., *Cell Metab*, 2016; Tarrago et al., *Cell Metab*, 2018). Nicotinamide phosphoribosyltransferase (NAMPT), a key enzyme in NAD⁺ biosynthesis, is notably affected by high-fat diets, leading to reduced NAD⁺ biosynthesis (Yoshino et al., *Cell Metab*, 2011). Mice with an adipocyte-specific deletion of NAMPT exhibit lower NAD⁺ levels in their fat tissues, suffer from multi-organ insulin resistance, and have

impaired adipose tissue function. Remarkably, these issues can be ameliorated with NMN supplementation (Stromsdorfer et al., Cell Rep, 2016).

These studies collectively provide compelling evidence that targeting NAD⁺ metabolism can be an effective strategy against metabolic diseases. We have included the information in our revised manuscript (Discussion).

References

- Bai, P., Cantó, C., Oudart, H., Brunyánszki, A., Cen, Y., Thomas, C., Yamamoto, H., Huber, A., Kiss, B., Houtkooper, R.H., Riekelt H., *et al.* (2011). PARP-1 Inhibition Increases Mitochondrial Metabolism through SIRT1 Activation. *Cell Metabolism* *13*, 461-468.
- Barbosa, M.T., Soares, S.M., Novak, C.M., Sinclair, D., Levine, J.A., Aksoy, P., and Chini, E.N. (2007). The enzyme CD38 (a NAD glycohydrolase, EC 3.2.2.5) is necessary for the development of diet-induced obesity. *FASEB J* *21*, 3629-3639.
- Camacho-Pereira, J., Tarrago, M.G., Chini, C.C.S., Nin, V., Escande, C., Warner, G.M., Puranik, A.S., Schoon, R.A., Reid, J.M., Galina, A., *et al.* (2016). CD38 Dictates Age-Related NAD Decline and Mitochondrial Dysfunction through an SIRT3-Dependent Mechanism. *Cell Metab* *23*, 1127-1139.
- Canto, C., Houtkooper, R.H., Pirinen, E., Youn, D.Y., Oosterveer, M.H., Cen, Y., Fernandez-Marcos, P.J., Yamamoto, H., Andreux, P.A., Cettour-Rose, P., *et al.* (2012). The NAD(+) precursor nicotinamide riboside enhances oxidative metabolism and protects against high-fat diet-induced obesity. *Cell Metab* *15*, 838-847.
- Cross M, M.I., Wedel A, Renkawitz R. (1988). Mouse lysozyme M gene: isolation, characterization, and expression studies. *Proc Natl Acad Sci U S A* *85(17):6232-6*.
- Klockars M, and EF., O. (1974). Localization of lysozyme in normal rat tissues by an immunoperoxidase method. *J Histochem Cytochem.* . *J Histochem Cytochem* *22(3):139-46*.
- Stromsdorfer, K.L., Yamaguchi, S., Yoon, M.J., Moseley, A.C., Franczyk, M.P., Kelly, S.C., Qi, N., Imai, S., and Yoshino, J. (2016). NAMPT-Mediated NAD(+) Biosynthesis in Adipocytes Regulates Adipose Tissue Function and Multi-organ Insulin Sensitivity in Mice. *Cell Rep* *16*, 1851-1860.
- Tarrago, M.G., Chini, C.C.S., Kanamori, K.S., Warner, G.M., Caride, A., de Oliveira, G.C., Rud, M., Samani, A., Hein, K.Z., Huang, R., *et al.* (2018). A Potent and Specific CD38 Inhibitor Ameliorates Age-Related Metabolic Dysfunction by Reversing Tissue NAD(+) Decline. *Cell Metab* *27*, 1081-1095 e1010.
- Yoshino, J., Mills, K.F., Yoon, M.J., and Imai, S. (2011). Nicotinamide mononucleotide, a key NAD(+) intermediate, treats the pathophysiology of diet- and age-induced diabetes in mice. *Cell Metab* *14*, 528-536.
- Zhang, Q., Pan, Y., Zeng, B., Zheng, X., Wang, H., Shen, X., Li, H., Jiang, Q., Zhao, J., Meng, Z.X., *et al.* (2019). Intestinal lysozyme liberates Nod1 ligands from microbes to direct insulin trafficking in pancreatic beta cells. *Cell Res* *29*, 516-532.

Re: mSystems01214-23R1 (**Intestinal Lysozyme1 Deficiency Alters Microbiota Composition and Impacts Host Metabolism Through the Emergence of NAD⁺-Secreting ASTB Qing110 Bacteria**)

Dear Prof. Zhihua Liu:

Your manuscript has been accepted, and I am forwarding it to the ASM production staff for publication. Your paper will first be checked to make sure all elements meet the technical requirements. ASM staff will contact you if anything needs to be revised before copyediting and production can begin. Otherwise, you will be notified when your proofs are ready to be viewed.

Featured Image Submissions: If you would like to submit a potential Featured Image, please email a file and a short legend to msystems@asmusa.org. Please note that we can only consider images that (i) the authors created or own and (ii) have not been previously published. By submitting, you agree that the image can be used under the same terms as the published article. Image File requirements: TIF/EPS, 7.5 inches wide by 8.25 inches tall (at least 2,250 pixels wide by 2,475 pixels tall), minimum 300 dpi resolution (600 dpi preferred), RGB, and no figure elements, e.g., arrows or panel labels. The legend should be a short description of the image, 1-2 sentences recommended.

We recognize that the video files can become quite large, so to avoid quality loss ASM suggests sending the video file via <https://www.wetransfer.com/>. When you have a final version of the video and the still ready to share, please send it to mSystems staff at msystems@asmusa.org.

Sincerely,
Hongwei Zhou

Editor
mSystems

Reviewer #1 (Comments for the Author):

The authors have nicely addressed all my concerns. I think the paper is now suitable for mSystems.

Reviewer #2 (Comments for the Author):

authors have addressed all concerns.